# Hydroxytyrosol Interference with Inflammaging via Modulation of Inflammation and Autophagy

**DOI:** 10.3390/nu15071774

**Published:** 2023-04-05

**Authors:** Francesca Velotti, Roberta Bernini

**Affiliations:** 1Department of Ecological and Biological Sciences (DEB), University of Tuscia, Largo dell’Università, 01100 Viterbo, Italy; 2Department of Agriculture and Forest Sciences (DAFNE), University of Tuscia, Via S. Camillo de Lellis, 01100 Viterbo, Italy; roberta.bernini@unitus.it

**Keywords:** polyphenols, extra-virgin olive oil, olive vegetation waste, hydroxytyrosol, aging, inflammaging, inflammation, autophagy, cytokines, age-related diseases

## Abstract

Inflammaging refers to a chronic, systemic, low-grade inflammation, driven by immune (mainly macrophages) and non-immune cells stimulated by endogenous/self, misplaced or altered molecules, belonging to physiological aging. This age-related inflammatory status is characterized by increased inflammation and decreased macroautophagy/autophagy (a degradation process that removes unnecessary or dysfunctional cell components). Inflammaging predisposes to age-related diseases, including obesity, type-2 diabetes, cancer, cardiovascular and neurodegenerative disorders, as well as vulnerability to infectious diseases and vaccine failure, representing thus a major target for anti-aging strategies. Phenolic compounds—found in extra-virgin olive oil (EVOO)—are well known for their beneficial effect on longevity. Among them, hydroxytyrosol (HTyr) appears to greatly contribute to healthy aging by its documented potent antioxidant activity. In addition, HTyr can modulate inflammation and autophagy, thus possibly counteracting and reducing inflammaging. In this review, we reference the literature on pure HTyr as a modulatory agent of inflammation and autophagy, in order to highlight its possible interference with inflammaging. This HTyr-mediated activity might contribute to healthy aging and delay the development or progression of diseases related to aging.

## 1. Introduction

Aging is a complex physiological condition, derived by the time-dependent lowering of cell functions and dysregulation of molecular mechanisms underlying many cellular homeostatic processes [1].

A well-established hallmark of aging is the chronic state of immune activation causing low-grade inflammation in several tissues and organs, referred to as inflammaging [2,3]. The mechanisms underlying inflammaging are not yet completely understood. However, continuous immune stimulation associated with cellular stress and genetic factors appears to play a key role in generating this condition [4,5,6]. It has been proposed that a major source of immune stimulation in aging consists of endogenous/self, misplaced or altered molecules resulting from aged damaged/dead cells and organelles, and cell debris which, recognized by innate immune receptors, induces activation of inflammatory signaling pathways [4]. A key role in inflammaging is played by the activation of innate immune cells (mostly recruited monocyte-derived macrophages and tissue-resident macrophages, such as microglia), associated with upregulation of innate immune receptors and increased activity of pro-inflammatory signaling pathways [4,6]. Particular emphasis has been given to the balance of macrophage polarization into classically pro-inflammatory M1 and alternatively anti-inflammatory M2 activated macrophages, which is critical for immune homeostasis [4]. In aging, continuous immune stimulation is also associated with a decline in macroautophagy (herein referred to as autophagy)—a lysosome-mediated degradation process appointed to remove unnecessary or dysfunctional cell components [7,8,9]. Autophagy can regulate the function of immune and non-immune cells, affecting their inflammatory response [10,11]. Decreased autophagic activity can contribute to inflammaging in different ways, such as the direct activation of inflammatory signaling pathways as well as the impairment of cellular housekeeping, leading to the accumulation of damaged cellular and protein components, which represent inflammatory stimuli promoting inflammation [12]. Such a systemic imbalance between increased immune stimulation and reduced autophagy represents a major driving force for frailty, leading to aged dysfunctional immune and non-immune cells, declined immune functions and thus vulnerability to infectious diseases and vaccine failure.

The systemic pro-inflammatory status in inflammaging results in the rise of multiple soluble inflammatory mediators, secreted by immune and non-immune cells in the tissues and serum of the elderly. The inflammatory biomarkers of inflammaging include cytokines such as IL-1β, IL-18, IL-6, TNF-α, IL-17, IFN-γ, macrophage migration inhibitory factor (MIF), chemokines such as C-X-C motif ligand (CXCL) 8/IL-8, CXCL10/interferon-induced protein 10 (IP-10), CXCL13, C-C motif ligand (CCL)2/monocyte chemoattractant protein 1 (MCP-1) and CCL5/regulated on activation normal T cell expressed and secreted chemokine (RANTES), as well as other proteins such as C-reactive protein (CRP) [13,14,15,16,17]. Increased levels of these inflammatory mediators in the elderly represent a highly significant risk factor for most age-related diseases, including obesity, type 2 diabetes, osteoporosis, cancer, cardiovascular and neurodegenerative diseases [18,19,20,21,22]. For example, TNF-α, IL-1β and IL-6 cytokine levels are increased in the parkinsonian brain and serum [22,23,24,25], while the CCL2 chemokine level is raised in neuroinflammation associated to neurodegenerative diseases [26,27]. CXCL10 has been shown to be a biomarker for heart failure development and left ventricular dysfunction [28]. In addition, aging can also be associated with the dysregulated production of anti-inflammatory mediators, including cytokines such as IL-10 and IL-4 [21,29,30]. Therefore, inflammaging constitutes the basis for the onset or progression of chronic inflammatory diseases, which accelerate and propagate the aging process locally and systemically [31,32,33]. Consequently, inflammaging can be considered a major target for anti-aging strategies.

The Mediterranean diet (MD), based on fruit and vegetable intake, has been associated with a low occurrence of the diseases related to aging [34]. Daily consumption of extra-virgin olive oil (EVOO) is highly recommended for the beneficial role of polyphenols found in *Olea europaea* L. on human health [35,36,37,38]. The main phenolic compound is hydroxytyrosol (HTyr) (Figure 1). The central role of HTyr and its derivatives in the prevention of oxidative stress has been recognized by the European Food Safety Authority (EFSA), which recommends a daily intake of 20 g of EVOO containing at least 5 mg of these polyphenols [39].

HTyr is an amphiphilic compound; then, during olive oil processing, a fair amount spills into olive vegetation waste (OVW), such as olive oil wastewaters and olive pomace, which represents valuable sources from which to recover HTyr according to the “circular economy” model [40]. In recent times, HTyr has also been found in red wine, although with a lower concentration than in EVOO [41].

Our review is focalized on the potential effect of HTyr in inflammaging. In particular, we reference to literature investigating the role of purified HTyr as a modulatory agent of inflammation and autophagy in order to highlight its possible beneficial interference with inflammaging, an activity that might contribute to healthy aging and delay the development or progression of diseases related to aging.

## 2. Hydroxytyrosol (HTyr)

HTyr is the main phenol present in EVOO. It is a small, catecholic compound exhibiting antioxidant, cardioprotective, anticancer, neuroprotective and anti-inflammatory activities [42,43,44,45,46,47]. Because of these interesting properties, it has a high therapeutic potential and is used as a supplement and preservative for cosmeceutical, nutraceutical and food applications.

Despite the biological relevance of HTyr, its biosynthesis is still not fully clear. According to Figure 1, the proposed pathway in *Rhodiola crenulate* starts from L-DOPA, which is decarboxylated to afford dopamine; dopamine is firstly deaminated to 3,4-dihydroxyphenylacetaldehyde and finally reduced to HTyr [48].

As olives ripen and during their processing, the amount of HTyr increases due to the hydrolysis of oleuropein, which produces glucose, elenolic acid and HTyr, according to Figure 2 [49]. The same reaction occurs after intake of oleuropein under gastric conditions [50] and by the action of the microbiota [51].

Being a hydrophilic compound, during the production of EVOO, HTyr splits between the oil and liquid wastes such as olive oil wastewater or in solid wastes such as olive pomace, depending on the partition coefficients. These wastes represent a valuable source of HTyr, from which it can be recovered according to the concept of the “circular economy” [40]. Membrane technologies are sustainable and efficient to produce HTyr-enriched extracts [52,53,54] and the content of HTyr can be increased, treating them with tyrosinase to convert tyrosol to HTyr [55]. However, these extracts contain other phenolic compounds, and the observed biological activities can be influenced by them. In recent times, HTyr has been identified in red wine [41] because of the action of yeasts, which convert tyrosine to HTyr during alcoholic fermentation [56].

Many in vitro and in vivo studies are focused on the ADMET (absorption, distribution, metabolism, excretion and toxicity) processes of HTyr responsible for its beneficial role on health [43,44]. The absorption occurs in the small intestine, but the efficiency of the process depends on the food matrix, age and gender. Experimental results have demonstrated that absorption is higher if HTyr is administered as EVOO [57] and in female rats [58]. If administered as red wine, the levels of HTyr increase in urine due to its production by ethanol and dopamine through dopaminergic pathways [59]. After the absorption, HTyr reaches its highest plasma concentration in 5–30 min, and it is distributed among the kidney, liver, muscle, and brain. These peculiar properties, along with the strong antioxidant effect, candidate HTyr as a dopaminergic neuronal protector. The metabolism of HTyr is mainly related to the production of HTyr acetate (HTyr-Ac), homovanillyl alcohol and homovanillic acid, which were converted into the corresponding sulfate and glucuronide by conjugation reactions catalyzed by uridine 5-diphosphoglucuronosyl transferases and sulfotransferases. Some HTyr metabolites are depicted in Figure 2 [43]. Additionally, the excretion of HTyr depends on the food matrix [60] and a gender-dependent relationship was observed [58]. The main urinary HTyr metabolites are glucuronides.

To the best of our knowledge, the toxicity of pure HTyr has not yet been studied. However, the safety profile has been studied for Hidrox^®^, an HTyr-enriched extract obtained by a patented process. In vivo studies have demonstrated a No Observed Adverse Effect Level (NOAEL) of 500 mg/kg/day of HTyr [61]. At this concentration, HTyr has no adverse effects related to genotoxicity and mutagenicity [62]. This formulation has been designated as Generally Recognized as Safe (GRAS) and has been evaluated on oxidative stress, neuroinflammation, apoptosis and inflammasomes [63].

Purified HTyr can be obtained after chromatographic separations of EVOO or HTyr-enriched extracts or after chemical or enzymatic synthesis. Several synthetic procedures have been optimized in recent years using tyrosol and homovanillic alcohol as starting materials and efficient oxidative stoichiometric or enzymatic oxidants such as 2-iodoxybenzoic acid or tyrosinase [64,65,66].

Synthesis also offers the solution to improve the bioavailability of HTyr. In fact, several lipophilic derivatives have been prepared [67,68,69,70]. The most interesting one is HTyr-Ac depicted in Figure 2 [71]. Dominguez-Perles et al. carried out an in vivo experiment demonstrating the higher absorption of HTyr-Ac compared to HTyr through the corresponding metabolites obtained after oral administration [58]. Recently, the more lipophilic compound HTyr triacetate, called by the authors peracetylated HTyr (Per-HTyr) (Figure 3), was tested as an anti-inflammatory agent [72].

## 3. Inflammation, Inflammaging and HTyr

### 3.1. Inflammation and Inflammaging

Inflammation represents a crucial line of defense against pathogens. It is an evolutionarily conserved protective process designed to maintain organismal homeostasis against acute perturbations and serves as an adaptive response to infections and injuries [73]. However, chronic, systemic inflammation develops progressively with age in healthy individuals and contributes to organismal deterioration through a process termed inflammaging [5,74,75].

The major inflammatory cells involved in inflammaging are macrophages, including recruited monocyte-derived macrophages and tissue-resident macrophages such as microglia in the brain and spinal cord [4,76,77]. However, recent reports indicate also the involvement of other immune cells, such as T lymphocytes, in eliciting aging-related tissue inflammation [78]. For example, myocardial inflammaging is related to increased accumulation of dysfunctional CD4+T cells secreting massive amounts of inflammatory interferon (IFN)-γ [79,80]. Therefore, we can assume that, in inflammaging, both innate and adaptive immune cells are characterized by enhanced expression and secretion of pro-inflammatory mediators, often associated with reduced production of anti-inflammatory molecules.

The raise of pro-inflammatory mediators in inflammaging is the result of the increased activation of intracellular signaling molecules, such as those belonging to the mitogen-activated protein kinase (MAPK) pathway [81,82], as well as transcription factors, with a pivotal role played by the nuclear factor kappa-light-chain enhancer of activated B cells (NF-κB) [83,84,85]. The MAPK signaling pathway is involved in transmitting extracellular signals to the nucleus, playing critical regulatory roles in the production of pro-inflammatory cytokines. The MAPK family consists of the following three subfamilies: the extracellular signal-regulated kinases (ERKs; ERK1/2 and ERK5), the c-Jun N-terminal kinases (JNKs; JNK1-3) and the p38MAPK, which activates transcription factors leading to inflammation [81]. Therefore, MAPKs are important targets for the treatment of chronic inflammatory diseases [82]. Then, the transcription factor NF-κB is the master regulator of the expression of pro-inflammatory genes (including those encoding cytokines and chemokines), and NF-κB upregulation has been well documented in age-related diseases [86,87]. NF-κB activation is induced by cell stressor signals such as antigens, toll-like receptor (TLR) ligands and cytokines (e.g., tumor necrosis factor-TNF), which, via intermediate steps, lead to the engagement and activation of the inhibitor of the nuclear factor-κB (IκB) kinase (IKK) complex, which guides the phosphorylation of IκB, its ubiquitination and degradation. Once degraded, the remaining NF-κB dimer, composed of p65/p50 subunits, translocates to the nucleus and activates the transcription of inflammatory genes [81,83]. The MAPK pathway and NF-κB activations, tightly regulated by phosphorylation and ubiquitination, are responsible for the production of inflammatory cytokines such as TNF-α and IL-6. In addition, NF-κB translocation to the nucleus is responsible for the expression of IL-1β and IL-18 pro-forms that are processed and released in their active forms by another critical component of the inflammatory response, such as the nucleotide binding domain-like receptor pyrin domain containing the protein 3 (NLRP3) inflammasome [88]. NLRP3 is the most extensively studied inflammatory activator among the inflammasomes. Inflammasomes are a family of intracellular multi-protein complexes, belonging to the family of cytoplasmic innate immune receptors, which activate inflammatory caspases [88]. The NLRP3 inflammasome is mainly expressed by inflammatory innate immune cells such as monocytes/macrophages, dendritic cells and neutrophils. However, it can be also expressed by T lymphocytes and plasma cells, as well as by non-immune cells, including endothelial cells, fibroblasts, muscle cells, adipose progenitor cells and adipocytes. NLRP3 inflammasome formation, whose signaling events and way of assembly are not fully understood, is triggered by pathogen- or damage- associated molecular patterns, which are generated by endogenous stress, tissue damage or metabolic imbalances. Thus, once the NLRP3 protein complex has formed, it activates caspase-1, which initiates the processing and release of active IL-1β and IL-18 pro-inflammatory cytokines. Increased NLRP3 activity associated with increased IL-1β and IL-18 levels have been demonstrated in the elderly, and contribute to inflammaging [75,88,89,90]. Indeed, increased NLRP3 activation is associated with several age-related pathological conditions, including Alzheimer’s disease, atherosclerosis and type-2 diabetes [91,92], whereas NLRP3 inhibition extends health-spans and diminishes age-dependent degenerative conditions [90].

Another biological system implicated in inflammation and inflammaging is the microbiota, which is the community of bacteria, viruses, fungi and protozoa living in the human body and interacting with it. We refer not only to the gut microbiota but also the microbiota at different body sites, whose composition and diversity change according to age [93,94]. Age-related microbiota changes (referred to as age-related dysbiosis) modulate the immune response and produce inflammatory molecules, contributing to immunosenescence and inflammaging by long-term stimulation of inflammation. On the other hand, by-products of metabolic processes in microbiota, including some short-chain fatty acids, can play a role in inhibiting inflammation [93,94].

Altogether, inflammation in inflammaging can be the result of the imbalance between pro- and anti-inflammatory molecules and underlies several age-related diseases.

### 3.2. HTyr as Modulatory Agent of Inflammation

Polyphenols have been widely recognized as anti-inflammatory agents [95,96,97,98,99,100]. In addition, it has been shown by our and other’s laboratories that HTyr-enriched preparations, including those derived from EVOO and OVW, as well as multicomponent nutraceutical containing HTyr, can modulate several key processes related to inflammation [55,63,101,102,103,104,105,106,107,108,109]. However, these preparations, containing other phenolic compounds as well as HTyr, do not allow the identification of immunomodulatory activity mediated by the single phenol and do not exclude the potential synergistic effects for the presence of other compounds. Therefore, here we reference studies that investigate the in vitro (Table 1) and in vivo (Table 2) modulation of inflammation by purified HTyr. Moreover, we focus our attention to data concerning the HTyr capability of directly affecting the mentioned inflammatory biomarkers of inflammaging, referring to other reviews for its antioxidant activity [63,96,103,110,111,112,113,114,115,116].

#### 3.2.1. Evidence In Vitro

Considering the central role of macrophages in inflammation, we first reference research investigating the in vitro immunomodulatory activity of HTyr on experimental models of murine and human inflammatory monocytes/macrophages (Table 1). Treatment of the human monocyte/macrophage cell line THP-1 with HTyr (25, 50, and 100 μM) for 10 min, before stimulation with lipopolysaccharide (LPS) for 3 h, resulted in a dose-dependent reduction of TNF-α transcription [117]. Then, 30 min of HTyr (0.25, 0.5, 1 μM) pre-treatment of oxysterol mixture-stimulated human peripheral blood mononuclear cells (PBMCs) from healthy donors reduced the secretion of pro-inflammatory cytokines and chemokines, such as IL-1β, MIF, RANTES as well as MAPK signaling, analyzed as p38MAPK and JNK phosphorylation [118]. In addition, HTyr (10, 20, 40, 80 μM) exerted anti-inflammatory effects on LPS-stimulated RAW264.7 murine macrophages, in that it downregulated LPS-induced expression of IL-1β and TNF-α, as well as suppressed NF-κB phosphorylation and activation [119]. Another study illustrated the high anti-inflammatory activity of Per-HTyr observed at lower doses than that used for HTyr. In fact, HTyr 50 (μM) and Per-HTyr (12.5, 25, 50 μM) significantly reduced the production of pro-inflammatory cytokines such as IL-1β, TNF-α, IL-6, IFN-γ and IL-17 by ex vivo murine peritoneal macrophages stimulated with LPS for 18 h. Both polyphenols also inhibited the activation of the signal transducer and activator of transcription 3 (STAT3), an important transcription factor involved in the regulation of immune response, inflammation and ageing [120]. Moreover, Per-HTyr suppressed the activation of the non-canonical NLRP3 inflammasome (which has the same effect as the canonical NLRP3, but is mediated by caspase-11 instead of caspase-1), decreasing thus the IL-18 pro-inflammatory cytokine level [72]. In another study, 12-h HTyr (50, 100 μM) pre-treatment of 24-h LPS-induced RAW 264.7 cells decreased pro-inflammatory M1/CD11c+ macrophages and increased anti-inflammatory M2/CD206+ macrophages, while reducing mRNA and protein levels of TNF-α, IL-1β, IL-6 pro-inflammatory cytokines and raising IL-10 and IL-4 anti-inflammatory cytokines. These results were also associated with a decrease in ERK1/2 phosphorylation, indicating that HTyr is able to promote macrophage M2 polarization with the increase of anti-inflammatory cytokines through the inhibition of the MAPK signaling pathway [121]. However, in contrast to the anti-inflammatory effects of HTyr reported by different investigators so far, in a small number of studies, HTyr appeared to exert no effect or even promote inflammation. In fact, Bigagli et al. reported that co-treatment of LPS-stimulated RAW 264.7 with low doses (5 and 10 μM) of HTyr for 18 h produced no change in IL-1β and TNF-α gene expression [122]. Then, co-incubation of LPS-activated human monocytes with HTyr (50, 100 μM) for 24 h increased TNF-α production by inflammatory monocytes [123]. Accordingly, the expression of TNF-α was significantly increased in ex vivo murine peritoneal macrophages co-incubated with LPS and HTyr (80 μM) for 24 or 48 h. In this study, HTyr also increased LPS-dependent expression of the IL-10 gene, whereas it did not affect LPS-dependent NF-κB gene expression and NF-κB phosphorylation [124].

Moreover, we reference studies on the immunomodulatory activity of HTyr on experimental models of inflammatory microglia cells (Table 1), specialized macrophages of the central nervous system and principal orchestrators of neuroinflammation. Gallardo-Fernandez et al. showed that HTyr (1, 10, 25 and 50 μM) was able to reduce mRNA levels of pro-inflammatory mediators such as TNF-α, IL-1β, IL-6 and CXCL10 expressed by LPS-induced BV2 murine microglial cells. The analysis of the possible molecular mechanisms involved in the anti-inflammatory effect showed that HTyr decreased the phosphorylated forms of JNK1/2 and p38MAPK, thus reducing MAPK signaling, as well as inhibiting the activation of the NLRP3 inflammasome. In addition, HTyr treatment prevented LPS-induced translocation to the nucleus of NF-κBp65 [125]. Accordingly, another study revealed that HTyr (25, 50, 100 μM) significantly decreased the production of IL-1 β, TNF-α and IL-6 by BV2 microglia and primary microglia cells stimulated with LPS. Moreover, HTyr significantly reduced M1/CD86+ macrophages and increased M2/CD206+ macrophages, decreased phosphorylated ERK and NF-κBp65 levels in a dose-dependent manner and suppressed the expression of toll like receptor 4 (TLR4) in BV2 microglia. These results suggest that HTyr can suppress LPS-induced neuroinflammatory responses via modulation of M1/M2 microglia polarization and downregulation of TLR-4 mediated NF-κB activation and the ERK signaling pathway [126]. Then, a study investigated the ability of HTyr (20, 100 μM) to alleviate neuropathic pain, by using an experimental model of intervertebral disc degeneration (IVDD). TNF-α-stimulated primary human nucleus pulposus cells (HNPCs) were used to simulate the local inflammatory microenvironment observed in IVDD, and LPS was used to stimulate rat microglia cells. HTyr inhibited the secretion of TNF-α, IL-6, IL-1β and the activation of the NLRP3 inflammasome, while it also suppressed NF-κB activation and ERK phosphorylation. These results suggest that HTyr plays a protective role against IVDD and secondary neuropathic pain by inhibiting NF-κB and MAPK inflammatory pathways [127].

Finally, the anti-inflammatory activity of HTyr has also been investigated in experimental models of inflammatory non-immune cells (Table 1), such as senescent fibroblasts [128], keratinocytes [129,130] and colonic epithelial cells [131]. Menicacci et al. investigated the ability of HTyr to affect the inflammatory activity of senescent cells, which display increased secretion of growth factors, inflammatory cytokines and proteolytic enzymes. Chronic (4–6 weeks) pre-treatment with HTyr (1 μM) of pre-senescent MRC5 human lung fibroblasts, neonatal human dermal fibroblasts (NHDF) and TNF-α-stimulated NHDF reduced IL-6 production in all cellular experimental models. In addition, HTyr decreased NFκB protein levels and nuclear localization. These data suggest that the modulation of the senescence-associated inflammatory phenotype might be another important activity underlying the anti-inflammatory activity of HTyr in inflammaging [128]. Then, primary human keratinocytes were pre-treated with HTyr (12.5–100 μM) or HTyr-Ac (12.5–100 μM) for 30 min and then stimulated with the IL-1β or TLR3 ligand (Poly I:C). The expression of TNF-α, IL-6 and IL-8 inflammation-related genes was likewise inhibited by HTyr-Ac and HTyr. Mechanistically, these polyphenols counteracted IκB degradation and translocation of NF-κB to the nucleus, in particular to the critical binding site in the IL-8 promoter [129]. Another study showed that HTyr exerted a potent anti-inflammatory effect on inflammatory keratinocytes in an experimental model of psoriasis, a prevalent chronic inflammatory dermatosis. Indeed, HTyr (25, 50, 100 μM) decreased the expression of IL-6, IL-8 and TNF-α in HaCaT keratinocytes stimulated with a M5 cytokine cocktail (composed of TNF-α, oncostatin-M, IL-17A, IL-1α and IL-22) [130]. Lastly, a recent study demonstrated the anti-inflammatory effect of HTyr on a human model of colonic chemical carcinogenesis. Primary human colonic epithelial cells (HCoEpC) were pre-treated with 1 μM HTyr 45 min before their exposition to benzo[a]pyrene [B[a]P]. Interestingly, all the inflammatory effects induced by B[a]P could be counteracted by HTyr. In fact, HTyr reduced the release of pro-inflammatory cytokines, chemokines and growth factors, including IL-6, IL-8, CXCL13 and vascular endothelial growth factor (VEGF), and molecules able to recruit inflammatory monocytes and activate macrophages, which could contribute to inflammation, angiogenesis and tumorigenesis. Moreover, HTyr efficiently counteracted B[a]P-mediated ERK1/2 phosphorylation. These data suggest that HTyr can play an important role in preventing tumorigenesis by chemical carcinogens [131].

**Table 1 nutrients-15-01774-t001:** In vitro effects of HTyr on inflammation.

Treatment and Dose	Model	Effects	Ref.
Pre-treatment 25, 50, 100 μM HTyr	Human Macrophages: THP-1 + LPS	↓ TNF-α	[117]
Pre-treatment 0.25, 0.5, 1 μM HTyr	Human Mononuclear Cells: PBMCs + oxysterol mixture	↓ IL-1β, MIF, RANTES ↓ p38MAPK and JNK phosphorylation	[118]
Co-treatment 10, 20, 40, 80 μM HTyr	Murine Macrophages: RAW 264.7 + LPS	↓ IL-1β, TNF-α ↓ NF-κB phosphorylation	[119]
Pre-treatment 50 μM HTyr 12.5, 25, 50 μM Per-HTyr	Murine Macrophages: ex vivo peritoneal macrophages + LPS	↓ IL-1β, IL-6, TNF-α, IL-17, IFN-γ ↓ STAT3 activation ↓ IL-18 via non-canonical NLRP3 inflammasome	[72]
Pre-treatment 50, 100 μM HTyr	Murine Macrophages: RAW 264.7 + LPS	↓ M1, ↑ M2 macrophages ↓ IL-1β, IL-6, TNF-α ↑ IL-10, IL-4 ↓ ERK1/2 phosphorylation	[121]
Co-treatment 5, 10 μM HTyr	Human Peripheral Blood Monocytes: ex vivo monocytes + LPS	No change in IL-1β, TNF-α	[122]
Co-treatment 50, 100 μM HTyr	Murine Macrophages: RAW 264.7 + LPS	↑ TNF-α	[123]
Co-treatment 80 μM HTyr	Murine Macrophages: ex vivo peritoneal macrophages + LPS	↑ TNF-α, IL-10 no change in NF-κB expression and phosphorylation	[124]
Co-treatment 1, 10, 25, 50 μM HTyr	Murine Microglia: BV2 + LPS	↓ IL-1β, TNF-α, IL-6, CXCL10 ↓ JNK1/2 and p38MAPK phosphorylation ↓ NF-κBp65 translocation to the nucleus ↓ NLRP3 inflammasome	[125]
Pre-treatment 25, 50, 100 μM HTyr	Murine Microglia: BV2 + LPS primary microglia + LPS	↓ M1 ↑ M2 ↓ IL-1 β, TNF-α, IL-6 ↓ TLR-4 ↓ NF-κBp65 phosphorylation ↓ ERK1/2 phosphorylation	[126]
Co-treatment 20, 100 μM HTyr	Human Nucleus Pulposus Cells and Rat Microglia: primary HNPC + TNF-α microglia + LPS	↓ IL-1β, TNF-α, IL-6 ↓ NLRP3 inflammasome ↓ NF-κB activation ↓ ERK phosphorylation	[127]
Pre-treatment 1 μM HTyr for 4 weeks	Human Pre-senescent and Senescent Fibroblasts: MRC5 NHDF NHDF + TNF-α	↓ IL-6 ↓ NF-κB activation	[128]
Pre-treatment 12.5–100 μM HTyr 12.5–100 μM HTyr-Ac	Human Keratinocytes: primary keratinocytes + IL-1β primary keratinocytes + Poly I:C	↓ TNF-α, IL-6, IL-8 ↓ NF-κB activation and translocation to binding site in the IL-8 promoter	[129]
Pre-treatment 25, 50, 100 μM HTyr	Human Psoriatic Keratinocytes: HaCaT + M5 cytokine cocktail	↓ IL-6, IL-8, TNF-α	[130]
Pre-treatment 1 μM HTyr	Chemical Carcinogenesis in Human Primary Colonic Epithelial Cells: HCoEpC + B[a]P	↓ IL-6, IL-8, VEGF, CXCL13 ↓ ERK1/2 phosphorylation	[131]

Abbreviations: Per-HTyr: peracetylated hydroxytyrosol. HTyr-Ac: hydroxytyrosyl acetate. LPS: lipopolysaccharide. PBMC: peripheral blood mononuclear cells. TNF: tumor necrosis factor. IL: interleukin. MIF: macrophage migration inhibitory factor. RANTES: regulated on activation normal T-cell expressed and secreted. CXCL: C-X-C motif ligand. VEGF: vascular endothelial growth factor. TLR: toll-like receptor. NF-κB: nuclear factor-κB. MAPK: mitogen-activated protein kinase. STAT: signal transducer and activator of transcription. ERK: extracellular signal-regulated kinase. JNK: c-Jun N-terminal kinase. HNPC: human nucleus pulposus cells. MRC5: human fibroblasts from fetal lung tissue. NHDFs: neonatal human dermal fibroblasts. B[a]P: Benzo[a]pyrene. ↓ Decrease. ↑ Increase.

#### 3.2.2. Evidence In Vivo

The modulation of inflammation by HTyr has also been investigated in vivo, in inflammatory experimental animal models. As for in vitro investigations, we start by reporting studies on the immunomodulatory effects of HTyr on innate immune cells, such as macrophages and microglia (Table 2). In an in vivo murine model of pristane-induced systemic lupus erythematosus, oral administration of 100 mg/kg (4 g/day for 6 month) HTyr was able to reduce IL-1β and IL-6 secreted by ex vivo LPS-stimulated splenocytes and macrophages. Studies on renal tissue, in the same experimental model, showed that HTyr also prevented the degradation of IκB, the nuclear translocation of NF-κBp65 and the phosphorylation of MAPK [132]. Then, in C57BL/6 mice with an LPS-induced acute liver injury, where infiltrated inflammatory macrophages play a critical role in liver destruction and dysfunction, HTyr (100 mg/kg orally, once a day daily, for 2 days) decreased inflammatory M1/CD11c+ macrophages and increased anti-inflammatory M2/CD206+ macrophages in the liver tissue, while it reduced liver TNF-α, IL-1β and IL-6 mRNA and augmented serum IL-10 and IL-4 protein levels. These data suggest that HTyr mitigated hepatic inflammation and injury through modulating macrophage-mediated inflammation [121]. In a model of atherosclerosis in apoE−/− mice, HTyr administration (10 mg/kg/day orally, for 16 weeks) significantly reduced the extent of aorta atherosclerotic lesions as well as serum CRP, TNF-α, IL-1β and IL-6 levels, while it increased IL-10. In addition, HTyr reduced the activation of inflammatory signaling molecules, such as p38MAPK phosphorylation and NFκ-B activation in the liver, suggesting that HTyr displayed anti-atherosclerotic action via modulating inflammation through the inhibition of inflammatory signaling molecules [133]. Accordingly, Pirozzi et al. showed that HTyr (10 mg/kg/day orally, for 5 weeks) exerted anti-inflammatory activity in the liver of rats with nonalcoholic fatty liver disease (NAFLD), by inhibiting liver TNF-α and IL-6 mRNA expressions [134]. Another study showed a potent anti-inflammatory activity of HTyr in a mouse model of systemic inflammation. In Balb/c mice pre-treated with HTyr (80 mg/kg/daily, for 2 or 5 days) and stimulated by intra-peritoneal injection of LPS, HTyr prevented the LPS-induced increase of TNF-α plasma levels [135].

Moreover, studies have also shown the capability of HTyr to inhibit neuroinflammation in vivo (Table 2). In this context, it is important to consider that HTyr is capable of crossing the blood brain barrier [136]. In mice pre-treated with HTyr (100 mg/kg by gavage, daily for 2 days) and administered with LPS, HTyr significantly damped the LPS-induced increase of IL-6, IL-1β and TNF-α mRNA levels in the brain [126]. Moreover, Yu et al. reported that intrathecal injection of HTyr (2 μL of HT 100 μM) alleviates neuropathic pain in rats undergoing chronic compression of the dorsal root ganglion. Notably, HTyr reduced the production of IL-1β, IL-6 and TNF-α inflammatory cytokines in the spinal dorsal horn. In addition, HTyr inhibited the activation of the ERK signaling pathway by reducing the levels of ERK phosphorylation in the spinal dorsal horn [127].

The anti-inflammatory activity of HTyr has also been investigated in inflammatory models of murine dextran sulfate sodium (DSS)-induced colitis [137,138]. A low dose (10 mg/kg) and high dose (50 mg/kg) of HTyr intervention significantly reduces the markers of DDS-induced colitis such as IL-6, IL-1β and TNF-α inflammatory cytokines and the NF-κB inflammatory pathway. Moreover, a high dose of HTyr increased the secretion of the IL-10 anti-inflammatory cytokine. It is also noteworthy that HTyr intervention transformed the gut microbiota, leading to a lower abundance of inflammation-related microbes (e.g., *Bacteroidaceae* and *Desulfovibrionaceae*) and a higher level of short-chain fatty acids producing bacteria (e.g., *Lachnospiraceae*, *Muribaculaceae*, *ASF356 and Colidextribacter*), indicating that the modulation of inflammation by HTyr is also mediated by its capability to modulate the microbiota [137]. In another study, it was found that HTyr (40 mg/kg/day orally for 14 days) exerted anti-inflammatory activity in murine DSS-induced ulcerative colitis by inhibiting NLRP3 inflammasome activation, through the suppression of the expression levels of NLRP3, caspase-1 and the apoptosis-associated speck-like protein containing a caspase recruitment domain (ASC) mRNA, thus decreasing IL-18 and IL-1β levels. Moreover, this study confirmed the capability of HTyr to modulate gut microbiota by exerting a shift from pathogenic bacteria (*Helicobacter*, *Staphylococcus*, *Desulfovibrio* and *Streptococcus*) to probiotics (*Lactobacillus*, *Lachnospiraceae* NK4A136 group, [*Ruminococcus*] torques group and *Roseburia*), and increasing the levels of short-chain fatty acids (total acid, acetate, propionate and butyrate) [138].

**Table 2 nutrients-15-01774-t002:** In vivo effects of HTyr on inflammation.

HTyr Treatment and Dose	Model	Effects	Ref.
100 mg/kg diet: 4 g/day orally for 6 months	Murine pristane-induced Systemic Lupus Erythematosus: ex vivo splenocytes+LPS ex vivo macrophages+LPS renal tissue	↓ IL-1β, IL-6 ↓ IκB degradation, p65NF-kB nuclear translocation ↓ MAPK phosphorylation	[132]
100 mg/kg/day orally for 2 days	Murine LPS-induced Acute Liver Injury: liver tissue liver macrophages serum	↓ M1 ↑ M2 macrophages ↓ IL-1 β, TNF-α, IL-6 ↓ IL-10, IL-4	[121]
10 mg/kg/day orally for 16 weeks	ApoE−/− Mice Atherosclerosis: blood heart tissue liver tissue	↓ IL-1β, TNF-α, IL-6, CRP ↑ IL-10 ↓ NF-kB activation ↓ p38MAPK phosphorylation	[133]
10 mg/kg/day orally for 5 weeks	Rats with Nonalcoholic Fatty Liver Disease: liver tissue	↓ TNF-α, IL-6	[134]
80 mg/kg/daily for 2 or 5 days	Murine LPS-induced Systemic Inflammation: plasma	↓ TNF-α	[135]
100 mg/kg/day orally for 2 days	Murine LPS-induced Brain Inflammation: brain tissue	↓ IL-6, IL-1 β, and TNF-α	[126]
2 μL of 100 μM injected intrathecally	Rat Chronic Compression of Dorsal Root Ganglion-induced Neuropathic Pain: spinal dorsal horn	↓ IL-1β, TNF-α, IL-6 ↓ ERK phosphorylation	[127]
10 and 50 mg/kg/day orally	Murine DSS-induced Colitis: colon tissue fecal samples	↓ IL-6, IL-1β, and TNF-α ↑ IL-10 ↓ NF-κB activation ↓ Inflammation-related microbes of gut microbiota	[137]
40 mg/kg/day orally for 14 days	Murine DSS-induced Colitis: colon tissue fecal samples	↓ IL-18 and IL-1β via ↓ NLRP3 inflammasome activation ↓ inflammation-related microbes of gut microbiota	[138]

Abbreviations: LPS: lipopolysaccharide. IL: interleukin. TNF: tumor necrosis factor. DSS: dextran sodium sulfate. NF-κB: nuclear factor-κB. MAPK: mitogen-activated protein kinase. ERK: extracellular signal-regulated kinase. ↓ Decrease ↑ Increase.

## 4. Autophagy, Inflammaging and HTyr

### 4.1. Autophagy and Inflammaging

Autophagy is an evolutionary conserved, highly regulated cellular process, responsible for the removal via lysosomal degradation of damaged cell material, such as misfolded proteins and unnecessary or dysfunctional organelles [139]. It can also regulate cellular energy balance by triggering energy production from its own components during nutrient starvation [140]. The autophagic process has a housekeeping role under physiological conditions and an adaptive, cytoprotective role under stress, controlling thus cellular and tissue homeostasis [141,142]. The autophagic process, its response to stress and its regulation has been described in detail elsewhere [143]. Briefly, autophagy is a dynamic process that is controlled by various signaling molecules, including the two main regulators such as the mammalian/mechanistic target of rapamycin (mTOR) and the 5′ adenosine monophosphate-activated protein kinase (AMPK), which act as autophagic inhibitor and activator, respectively [144,145]. mTOR is a serine–threonine kinase that senses cellular nutrient levels through its connection with different signaling pathways, including the MAPK/ERK1/2 pathway and the phosphatidylinositol 3-kinase (PI3K)/Ak strain transforming (Akt; also known as protein kinase B) pathway, which establish the PI3K/Akt/mTOR axis. Regarding the latter, PI3K, activated by many types of cellular stimuli, phosphorylates the phosphatidylinositol 4,5-bisphosphate (PIP2) to phosphatidylinositol (3,4,5)-trisphosphate (PIP3) that recruits and activates Akt via phosphorylation. Phosphorylated Akt, as a serine/threonine kinase, is able to stimulate mTORC1, which can directly activate a variety of cellular effectors, including the eukaryotic translation initiation factor 4E-binding protein 1 (4E-BP1), to inhibit autophagy [81,144]. The autophagic driver AMPK is a serine/threonine protein kinase and a highly conserved energy sensor, which can be activated by allosteric regulation or by upstream kinases. Activated AMPK phosphorylates integrated signaling networks to promote autophagy directly (by the activation of transcription factors) or indirectly (by the inhibition of the mTORC1 pathway) [146]. Other signaling molecules involved in autophagic regulation are sirtuins (SIRT), a family (SIRT1-7) of nicotinamide adenine dinucleotide (NAD^+^)-dependent enzymes, which play roles in diverse cellular processes including autophagy, senescence and cell survival [147]. Among them, SIRT1 and SIRT6, NAD+-dependent deacetylases belonging to class III histone/protein deacetylases and members of the silent information regulator 2 (Sir2) family, are predominantly localized in the nucleus and share functional similarities [148,149]. They play essential roles in promoting autophagy. SIRT1 can promote autophagy both directly, by interacting with autophagy related molecules (ATGs), and indirectly, by interacting with upstream regulators of autophagy, such as those activating AMPK or inhibiting the Akt/mTOR pathway [150]. SIRT6 can induce autophagy via inhibition of Akt signaling, thus inhibiting the Akt-mTOR pathway [151].

The most common markers used for monitoring autophagy are two proteins, the microtubule-associated protein 1 light chain 3 (LC3) and the sequestosome-1 (SQSTM1)/p62 [152]. LC3 is recruited from the cytosol and associates with the phagophore early in autophagy; this localization serves as a general marker for autophagic membranes and for monitoring the process in its development. The SQSTM1/p62 protein is an ubiquitin-binding scaffold protein, which co-localizes with cargo destined to be degraded by autophagy; itself is degraded in autolysosomes and serves as an index of autophagic degradation.

It is now well known that autophagy contributes to the extension of longevity and prevents age-related pathologies [142,153]. Among the beneficial effects on age-related processes, autophagy orchestrates the differentiation and metabolic state of innate and adaptive immune cells, inducing immune response improving effects [154,155,156]. In particular, autophagy plays a pivotal role in both the prevention of aging in immune and non-immune cells and the suppression of inflammaging [10,11,142,153]. Indeed, autophagy prevents inflammation by downregulating inflammation-related signals, reducing inflammatory cytokine expression and promoting apoptotic corpse clearance [10,11]. Autophagy can suppress inflammatory reactions not only directly, but also indirectly through its cytoprotective activity, by increasing cytoplasmic turnover that antagonizes the degradation of organelles and macromolecular complexes, thus ameliorating immune system homeostasis. Although the regulatory mechanisms for autophagy in inflammation are complex and remain to be fully elucidated, the balance between mTOR and AMPK activation has a central role in innate immune cell homeostasis and function [155,157]. It has been demonstrated that downregulation of autophagy in macrophages leads to inflammation. Autophagy can regulate the metabolic state of macrophages, in that mTOR activation, reducing the autophagic flux, induces a proliferative and pro-inflammatory M1 phenotype in macrophages, whereas AMPK activation, driving autophagy, promotes the function of anti-inflammatory M2 macrophages [141]. In addition, AMPK can inhibit the inflammatory response by inhibiting the pro-inflammatory NF-κB pathway in immune and non-immune cells [158]. Furthermore, enabling the removal of damaged cell material, autophagy also regulates the intracellular danger signal-sensing multiprotein platform such as the NLRP3 inflammasome, inhibiting thus the activation of IL-1β and IL-18 inflammatory cytokines [159,160]. Finally, autophagy regulates also the adaptive immune response, such as T lymphocyte function, including their metabolism, survival, development, proliferation, differentiation and aging. In particular, autophagy plays a pivotal role in regulating the function of T helper lymphocytes, regulating their secretion of cytokines in chronic inflammatory diseases [161].

Unfortunately, the aging process is associated with a decrease in autophagy, which further contributes to inflammaging, senescence and aging itself [153,158]. Decreased autophagy has been linked to a wide range of age-related inflammatory diseases, including cardiomyopathy, neurodegenerative disorders and cancer [141,142,162]. Therefore, compounds capable of promoting autophagy have recently received great attention for the treatment of age-related diseases [12].

### 4.2. HTyr as a Modulatory Agent of Autophagy

Polyphenols have recently been recognized as agents capable of modulating autophagy, improving thus cell homeostasis and function [163,164,165,166,167,168,169,170,171]. As for investigations concerning the modulation of inflammation by HTyr, most laboratories, including ours, performed investigations on the modulation of autophagy using HTyr-enriched preparations, which do not allow the identification of the effect mediated by the single HTyr compound [55,172,173]. Therefore, here we reference investigations that used purified HTyr to analyze its in vitro (Table 3) and in vivo (Table 4) capability to regulate the inflammaging markers through modulation of autophagy.

#### 4.2.1. Evidence In Vitro

Two studies exist indicating the capability of HTyr to affect inflammation and inflammaging markers by modulating autophagy in experimental models of chondrocytes stimulated by different inflammatory stimuli [174,175] (Table 3). Zhi et al. demonstrated that HTyr (25, 50, 100 μM) inhibited, in a dose-dependent manner, the levels of pro-inflammatory cytokines such as IL-1β and IL-6 as well as the chemokine MCP-1 in ex vivo rat primary chondrocytes stimulated with TNF-α. In addition, HTyr promoted autophagy and upregulated the expression of SIRT6 at the mRNA and protein level. Moreover, SIRT6 knockdown reduced the effect of HTyr on autophagy promotion by decreasing the expression of the LC3 autophagic marker. Furthermore, both the inhibition of autophagy by the autophagic inhibitor 3-methyladenine and SIRT6 knockdown decreased the suppressive effects of HTyr on the inflammatory response in TNF-α-induced chondrocytes. Overall, these findings indicate that HTyr inhibits TNF-α-induced inflammatory response in chondrocytes through regulating SIRT6-mediated autophagy, thus protecting chondrocytes from inflammation and suggesting potential beneficial effects of HTyr in osteoarthritis [174]. Then, Sun et al. investigated the anti-inflammatory effect of HTyr (75 μM) pre-treatment in rat primary chondrocytes stimulated with advanced oxidation protein products (AOPPs). The investigators found that HTyr increased the expression of LC3 and autophagy related (ATG) 5 and ATG 7 molecules, while it decreased the expression of p62, thus promoting autophagy in AOPP-stimulated chondrocytes. It was also demonstrated that autophagy was promoted through the SIRT1 pathway. Furthermore, HTyr inhibited the expression of IL-6 and TNF-α inflammatory cytokines at the mRNA and protein level, and this inhibition was suppressed by SIRT1 knockdown, showing that HTyr can inhibit the inflammatory response caused by stressed chondrocytes through the promotion of SIRT1-mediated autophagy [175].

A recent study investigated the mechanisms underlying the anti-inflammatory effect of HTyr in cardiovascular diseases, by analyzing the capability of HTyr to affect the inflammatory response in vascular adventitial fibroblasts (VAFs) through autophagy modulation (Table 3). Pre-treatment of TNF-α-stimulated rat primary VAFs (obtained from thoracic aorta) with HTyr (25, 50, 100 μM) promoted autophagy by increasing LC3 cell expression and the autophagic flux. In addition, HTyr upregulated SIRT1 mRNA and protein expression in TNF-α-stimulated VAFs, in a dose-dependent manner. It was also demonstrated that HTyr decreased Akt phosphorylation, and that the inhibition of both Akt and mTOR inhibited the promotion of autophagy by HTyr, indicating that HTyr modulated autophagy through SIRT1-mediated Akt/mTOR suppression. Furthermore, HTyr inhibited the TNF-α-induced inflammatory response in VAFs by inhibiting the secretion of IL-1 β and the expression of IL-6 and MCP-1 mRNA, while SIRT1 knockdown decreased both autophagy and inflammatory inhibition by HTyr. Therefore, these findings indicate that HTyr inhibits the inflammatory response produced by VAFs via the promotion of autophagy through SIRT1-mediated Akt/mTOR pathway suppression [176].

Another study analyzed the autophagic modulatory activity of HTyr in a human model of colonic chemical carcinogenesis (Table 3). HTyr (1 μM) pre-treatment of HCoEpC cells exposed to [B[a]P] restored B[a]P-mediated reduction in cell autophagy by increasing LC3 and reducing SQSTM1/p62 expression levels. HTyr also counteracted the phosphorylation of 4EBP1 (a downstream target of mTOR), thus counteracting mTOR activation. The modulation of autophagy by HTyr was associated with the inhibition of the release of the IL-6 pro-inflammatory cytokine, IL-8 and CXCL13 chemokines, and VEGF by HCoEpC cells exposed to [B[a]P]. These results suggest that HTyr might exert tumor suppression through the promotion of autophagy, thus playing an important role in preventing tumorigenesis by chemical carcinogens [131].

Lastly, Chen et al. explored the potential protective actions of HTyr-promoted autophagy in a rat model of stress-induced alopecia (Table 3). HTyr (75 μM) pre-treatment of H_2_O_2_-exposed rat primary dermal papilla cells promoted autophagy, inhibited mRNA and protein levels of IL-6 and TNF-α and enhanced the levels of hair growth factors such as the fibroblast growth factor (FGF) and platelet-derived growth factor (PDGF). By the use of the chloroquine autophagic inhibitor, it was also found that the anti-inflammatory effect was dependent on the regulation of autophagy by HTyr. These results indicate that HTyr is capable of significantly reducing stress-induced inflammation and that this effect is mediated by its capability to promote autophagy [177].

#### 4.2.2. Evidence In Vivo

The in vivo regulation of inflammation and inflammaging markers through the modulation of autophagy by HTyr has been investigated in two inflammatory experimental animal models [178,179] (Table 4). Yang et al. investigated the potential modulatory effect of HTyr on autophagy in a murine model of LPS-induced acute lung injury. Male BALB/c mice, challenged with intranasal instillations of LPS, were treated with HTyr (100 mg/kg) intragastrically 1 h prior to LPS exposure. Twenty-four hours later, lung and bronchoalveolar lavage fluid samples were obtained and analyzed for inflammatory and autophagic markers. LPS-driven release of TNF-α, IL-1β, IL-6, IL-10 and MCP-1 was strongly suppressed by HTyr. In addition, LPS-stimulated SIRT1 inhibition, MAPK phosphorylation and autophagy suppression were all abolished by HTyr administration. HTyr treatment significantly decreased pulmonary edema and inflammatory cell infiltration into lung tissues as well as inflammatory cell levels in bronchoalveolar fluid, suggesting that the protective effect of HTyr on lung inflammation may be attributed to the promotion of autophagy, which is likely associated with the activation of SIRT and MAPK signaling pathways [178]. Then, Nardiello et al. indicated the capability of HTyr to inhibit inflammation in a murine model of Alzheimer’s disease through the promotion of autophagy (Table 4). Indeed, in 4-month-old Tg (transgenic) CRND8 (overexpressing mutant human amyloid precursor protein) mice, treated for 8 weeks with a low-fat diet supplemented with HTyr (50 mg/kg of diet), a marked reduction of TNF-α expression in hippocampal areas associated with a strong promotion of autophagy in brain tissues was found, as indicated by the bright LC3 staining in the neuronal cell bodies and processes of neurons in the parietal cortex. These results were associated with a significant improvement of murine cognitive status and the reduction of plaque area and number of the amyloid aggregate of Aβ42 peptide and its pyroglutamated 3-42 derivative (pE3-Aβ) in the cortex and in the hippocampus, suggesting that HTyr mediates neuroprotection by modulating autophagy [179].

## 5. Discussion and Conclusions

In this review, we reference studies investigating the capability of purified HTyr to counteract the rise of multiple inflammatory mediators of inflammaging, by modulating inflammation and autophagy.

Encouraging results have been obtained concerning the modulation of inflammation by the administration of purified HTyr in different experimental models of inflammation in vitro and in vivo, indicating that HTyr can interfere with inflammaging by modulating inflammation. In fact, HTyr decreases inflammatory biomarkers of inflammaging, such as pro-inflammatory cytokines, chemokines and acute-phase proteins, by affecting different steps of the inflammatory process in vitro and in vivo. Indeed, HTyr can downmodulate the expression of the TLR-4 innate immune receptor (Table 1). HTyr also inhibits the activation of different intracellular signaling molecules involved in the inflammatory response, including ERKs, JNK and p38 MAPK, which belong to the MAPK pathway (Table 1 and Table 2). HTyr can suppress the activation of NF-κB, the master regulator of inflammatory gene transcription, as well as the NLRP3 inflammasome, a critical component of the inflammatory response (Table 1 and Table 2). Of note, HTyr can also exert anti-inflammatory activity by modulating the composition and diversity of microbiota, in particular the gut microbiota (Table 2). Furthermore, the inflammation modulating activity of HTyr has been illustrated in vitro in different inflammatory cell experimental models, including innate immune cells such as macrophages and microglia, as well as non-immune cells such as senescent fibroblasts, keratinocytes and colonic epithelial cells (Table 1). However, contrasting results have been reported regarding the capability of HTyr of inhibiting inflammation mediated by monocytes/macrophages (Table 1). We think that the diverse results might be related to the different experimental conditions and protocols (i.e., monocyte/macrophage experimental models, HTyr dose, times of HTyr and LPS incubations, times in which samples were taken) used by independent investigators, while Pojero et al. suggested that it might depend on the timing of HTyr administration (i.e., before, during or after the incubation with LPS inflammatory stimuli) [97]. Further experiments are needed to explain these contrasting results. The inflammation modulating activity of HTyr has also been illustrated in vivo in different experimental animal models of inflammatory diseases, by analyzing ex vivo and in vivo macrophages, blood samples, as well as renal, liver, heart, brain and colon tissues (Table 2). Interestingly, the capability of HTyr to affect M1/M2 macrophage/microglia polarization in vitro and in vivo has also been reported, highlighting the capability of HTyr to increase the activation of the M2 anti-inflammatory macrophage phenotype associated with the rise of anti-inflammatory cytokines (Table 1 and Table 2).

Promising results have also been reported concerning the possible interference of HTyr with inflammaging through the modulation of autophagy (Table 3 and Table 4). However, a small number of studies analyzed this issue, especially in in vivo experimental models (Table 4). It has been clearly shown that HTyr is capable of inhibiting the rise of inflammaging markers such as cytokines, chemokines and growth factors through the promotion of autophagy, by affecting the activation of regulatory molecules of the autophagic process in vitro and in vivo (Table 3 and Table 4). In fact, HTyr can upregulate autophagy by activating AMPK or inhibiting the Akt/mTOR pathway, via SIRT1 and SIRT6 modulation (Table 3 and Table 4). Furthermore, the capability of HTyr to decrease the expression of pro-inflammatory mediators through the promotion of autophagy has been observed in different experimental models, including arthritis, vasculitis and colonic chemical carcinogenesis in vitro, as well as acute lung injury and Alzheimer’s disease in vivo. However, although the capability of upregulating autophagy by HTyr-enriched preparations has been established in inflammatory immune cells such as macrophages [55], to the best of our knowledge, there is no study that has investigated the capability of pure HTyr to modulate autophagy in immune cells, including macrophages. Therefore, there is a need for studies on the capability of pure HTyr to interfere with inflammaging through modulating autophagy in innate and adaptive immune cell-mediated inflammatory responses.

Overall, our knowledge on inflammaging and its relationship with inflammation and autophagy is far from being complete, and there is still much to learn about the capability of pure HTyr to interfere with inflammaging via the modulation of inflammation and autophagy [180]. We consider that a certain number of critical issues, including the effective dosage and time of HTyr administration, have to be addressed before proceeding with clinical trials based on the administration of pure HTyr in elderly. In fact, the low bioavailability of orally ingested polyphenols, including HTyr, greatly limits their therapeutic use, particularly in those organs distant from the gastrointestinal tract. In addition, most of the in vitro and in vivo studies performed so far suggest a possible preventive rather than therapeutic activity of HTyr against inflammaging, in that in most studies, HTyr was administered before rather than after the inflammatory stimuli.

In conclusion, we consider that the promising results, emerged from preclinical investigations, strengthen a need for further studies aimed to better characterize the interference of HTyr with inflammaging to predict the possible effective clinical use of this compound in aging and age-related disorders.

## Data Availability

No data was used for the research described in the article.

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
