# Peer review of "Hydroxytyrosol Interference with Inflammaging via Modulation of Inflammation and Autophagy"

_nutrients, 2023, doi:10.3390/nu15071774_

Round 1
Reviewer 1 Report
Dear Authors of "Hydroxytyrosol Interference with Inflammaging via Modulation of Inflammation and Autophagy":
The proposed topic is quite interesting and innovative. English language is good and only minor style and spell errors have been detected. However, I find some points that need probable revision interventions:
Major issues:
1. To include some clinical studies although Hydroxytyrosol is administered with other antioxidant compounds, e.g.: Mosca A, Crudele A, Smeriglio A, Braghini MR, Panera N, Comparcola D, Alterio A, Sartorelli MR, Tozzi G, Raponi M, Trombetta D, Alisi A. Antioxidant activity of Hydroxytyrosol and Vitamin E reduces systemic inflammation in children with paediatric NAFLD. Dig Liver Dis. 2021 Sep;53(9):1154-1158. doi: 10.1016/j.dld.2020.09.021. Epub 2020 Oct 13. PMID: 33060043.
2. To include a graphical abstract to sum up all the biological processes and main molecular mechanisms regulated by Hydroxytyrosol to prevent and/or treat different pathologies included in the manuscript.
3. To explain in more detail the tissue expression of NLRP3 and how it is activated under different stressor stimuli
Minor issues:
1. To write "in vitro" and "in vivo" in italics in all the manuscript
2. The sentence "Moreover, Per-HTyr sup-274 pressed the activation of the non-canonical NLRP3 inflammasome (which inhibits 275 caspase-11), decreasing thus the IL-18 pro-inflammatory cytokine level" is not well expressed. Please, reformulate it.
3. Line 361: The word "p65NF-B" is not well-written.
4. Line 507: "4.2. HTyr as a Modulatory Agent of Autophagy"
5. Table 3: Chemical Carcinogenesis in Human Primary Colonic Epithelial Cells: HCoEpC + B[a]P: autophagy via????
6. The title of reference 72 is duplicated
7. To change mTORC1 instead of mTOR in both lines 447 and 453
Reviewer 2 Report
The manuscript is well written and adds knowledge about the use of phenolic compounds in extra virgin olive oil in the human diet as a modulator of inflammation and autophagy, in order to highlight their possible interference in inflammation. It is known that the phenolic compounds of extra virgin olive oil (EVOO) are recognized for their beneficial health effects associated with the promotion of longevity.
The manuscript brings together several bibliographies published in indexed journals, which describe relevant results, which makes the manuscript robust.
I did not identify any conceptual error or any other error that compromises the quality of the manuscript. I suggest a review in English, which contains minor errors.
